# Longitudinal associations between poor reading skills, bullying and victimization across the transition from elementary to middle school

Tiina Turunen[1]*, Elisa Poskiparta[1], Christina Salmivalli[1,2], Pekka Niemi[1], Marja-Kristiina Lerkkanen[3,4]

**1** Department of Psychology and Speech-Language Pathology, University of Turku, Turku, Finland, **2** Shandong Normal University, Jinan, China, **3** Department of Teacher Education, University of Jyväskylä, Jyväskylän, Finland, **4** University of Stavanger, Stavanger, Norway

* tmturu@utu.fi

## Abstract

Students with poor reading skills and reading difficulties (RDs) are at elevated risk for bullying involvement in elementary school, but it is not known whether they are at risk also later in adolescence. This study investigated the longitudinal interplay between reading skills (fluency and comprehension), victimization, and bullying across the transition from elementary to middle school, controlling for externalizing and internalizing problems. The sample consists of 1,824 students (47.3% girls, T1 mean age was 12 years 9 months) from 150 Grade 6 classrooms, whose reading fluency and comprehension, self-reported victimization and bullying, and self-reported externalizing and internalizing problems were measured in Grades 6, 7, and 9. Two cross-lagged panel models with three time-points were fitted to the data separately for reading fluency and comprehension. The results indicated that poorer fluency and comprehension skills in Grade 6 predicted bullying perpetration in Grade 7, and poorer fluency and comprehension skills in Grade 7 predicted bullying perpetration in Grade 9. Neither fluency nor comprehension were longitudinally associated with victimization. The effects of reading skills on bullying perpetration were relatively small and externalizing problems increased the risk for bullying others more than poor reading skills did. However, it is important that those who struggle with reading get academic support in school throughout their school years, and social support when needed.

**Data Availability Statement:** All relevant data are within the paper and its Supporting Information files.

## Introduction

Early adolescence is a period of change in many domains. Physical, cognitive, and social changes occur gradually and at individual pace, but the educational transition from elementary to middle school has a similar schedule for the entire peer group. Although for most adolescents acclimation to middle school is fast and successful [1–4], the transition can also be a stressful experience [5–8]. In middle school, the learning environment becomes more complex

**Funding:** Writing this article was supported by the INVEST Research Flagship, funded under the flagship scheme of the Academy of Finland (decision number: 320162), and another grant from the same funding agency (2013-2017, #268586).

**Competing interests:** The authors have declared that no competing interests exist.

due to both academic and social demands. Students from several elementary schools often move to a new, bigger school building with different teacher in each subject. This brings about social challenges, as the size and the structure of the peer groups change, and students are looking for their own niche in the new social structure [9, 10]. Prior transition, a major concern for elementary school students is being bullied in middle school [11–14].

Bullying is deliberate, repeated aggressive behavior that involves a real or perceived power imbalance between the bully and the victim [15, 16]. Transition to a new school can be an opportunity for students involved in bullying to form new, more positive relationships with their peers [17, 18]. On the other hand, this point in time may include further risks for peer problems in new environment. The percentage of students who are being bullied seems to decrease in higher grades in Finland as well as in other countries [19–25]. Although the overall bullying perpetration rate also decreases [9], studies have identified a group of students who report higher rates of bullying perpetration after the transition to middle school [10, 22, 26, 27], or even an increase in aggression and prevalence of bullies in middle school [9], as has also been observed in Finnish schools [24, 25]. Previous studies have shown that in middle school victimization seems to be more targeted than in elementary school: there are more bullies targeting fewer victims [i.e., 24]. These differences have been suggested to be at least partly due to differences between elementary and middle school characteristics, such as varying and more impersonal classes, various subject teachers, competition and social comparisons between peers, and teacher attitudes toward bullying [9].

Albeit being to some extent stressful for everyone, transition to middle school may lead to aggravation and adjustment problems especially for students with specific learning disabilities such as reading difficulties (RDs) and poor reading skills, since these students often come to middle school with a history of poor performance and social problems [i.e., 28]. In addition to being an academic impediment, RDs have been linked to social challenges, such as poor social skills and lack of social competence [29–31], low self-esteem [32], peer rejection [33, 34], as well as bullying involvement [35, 36]. These problems may already have existed throughout the elementary school, and transition to a new middle school adds to the burden. The present study investigates the interplay between reading skills, victimization and bullying across the transition from elementary to middle school in Finnish school context where students are 12 years old when they transit from elementary school (Grade 6) to middle school (Grade 7).

## Reading difficulties and bullying

RDs are among the most common learning difficulties affecting 3% to 17.5% of children and adults [37–39]. In the nationwide School Health Promotion study [25], 24% of the students in Grades 4 and 5, and 21.1% of the students in Grades 8 and 9 in Finland reported some or a lot of difficulties in reading in 2019. Furthermore, Kairaluoma et al. [40] identified with a screening test 7.4% of ninth grade students and 3.4% of vocational school students in their sample to be poor readers. This indicates that even in a language with a transparent orthography such as Finnish, reading difficulties persist long into adolescence, as has been reported in less transparent languages like English [41]. Whereas examining bullying among heterogeneous groups of students with a variety of learning problems is relatively common [e.g., 42–48], few studies have examined specific reading difficulties in association with bullying. However, there is some support for RDs and poor reading skills increasing the risk for both victimization and bullying others.

To begin with, struggling with the skill needed in school every day is likely to influence the overall experience of school, making it burdensome and aversive, and affecting emotions [e.g. 49–52] and behavior [53–56] at school. In interviews, children and adolescents with RDs have

reported victimization rates from about a third [57], to 50% [32], and up to 85% [58]. Even adults have frequently reported negative memories of victimization due to learning problems [e.g., 59]. Moreover, Turunen et al. [36] found in a nationally representative community sample of elementary and middle school students that over a third of those with self-reported RDs were involved in bullying as victims, bullies, or bully/victims, compared with approximately a fifth of students not reporting RDs. After controlling for gender, grade level, self-esteem, and difficulties in math, RDs specifically increased the risk of being victimized (viewed by peers as victims or bully/victims), but the association between RDs and bullying others (viewed by peers as bullies) was no longer significant.

Poor reading skills have also been related to bullying perpetration, although relevant studies are rare. Kaukiainen and his colleagues [60] found that learning difficulties, defined as difficulties in reading and writing, were associated with bullying, but not with victimization. In the study by Turunen et al. [35], poor readers with comorbid externalizing/internalizing problems were involved as bullies and bully/victims, but not as victims, in the beginning of elementary school. Skilled readers and struggling readers without externalizing/internalizing problems were not at elevated risk to be involved in bullying. The authors concluded that difficulties in reading are important indicators of bullying involvement among children in the beginning of elementary school only when occurring in tandem with externalizing/internalizing problems.

Previous studies have some limitations. Studies relying on interviews provide retrospective information about the phenomenon within samples of individuals with RDs only, without a comparison group that would make results more generalizable [i.e., 58, 59]. Turunen et al. [35] studied associations between word reading skills and bullying involvement in the beginning of elementary school, thus providing information only about young children and not taking into account reading comprehension. Although another study by Turunen et al. [36] included also adolescents by utilizing a community sample of elementary and middle school students, the results were cross-sectional and based on self-reports of reading problems instead of reading test scores. Therefore, it is not yet known whether and how poor reading skills are longitudinally related to bullying involvement in adolescence. Moreover, to our knowledge no previous study has examined reading comprehension in relation to bullying involvement.

## Transition to middle school for students with reading difficulties

To our knowledge, adjustment or bullying involvement across the transition to middle school among students with poor reading skills has not been studied before. Regarding academic adjustment, studies examining reading skills of general populations have revealed that during the transition from elementary to middle school, students' reading achievement may stall or even decline to levels below the elementary school achievement [61]. Also, a more general "dip" in the academic progress after transition has been reported [5, 62], although Akos et al. [63] found evidence of the transition effect as an interruption in academic growth rather than decline in the skills. The interruption was larger for vulnerable students, such as students receiving special education.

Although there is lack of research on middle school transition among students with RDs, a few studies have investigated the transition of students with heterogeneous Special Educational Needs (SEN). Compared with their peers, they have more concerns regarding bullying [1], even though their transition experience may not be less successful than that of their peers [1, 64]. However, most studies on adolescents with SEN report more victimization, adjustment and mental health problems, less social support, and academic challenges during the transitional year [1, 28, 65–69]. They also seem to have more internalizing and externalizing problems, and be rejected by their peers [68]. In addition to SEN, lower academic achievement has

been related to poorer transition to middle school [8, 70]. Interestingly, a study by Bailey and Baines [71] suggests that high levels of specific resilience factors (i.e., optimism and support) in primary school may leave SEN students less prepared for middle school. Researchers speculate that these students may underestimate the challenges they face, or may lack the skills to adapt because of previous overreliance on support, and thus experience difficulties in adjusting to their new schools.

Adjustment to middle school seems to involve developing a new set of social skills [72], which may be strenuous for students with RDs, since they often have challenges in social skills and competencies to begin with [29–31]. While the difficulties poor readers experience with studying and peers may not be new, they are likely to become more stressful as adolescence is a period of seeking greater autonomy, identity, and more intimate peer relationships [28]. On top of more complex social and learning environment, less support and warmth from teachers [73], and, at least in Finland, fewer special education services are available in middle school compared with elementary school [37]. Since poor reading skills are related to bullying involvement in younger students [35], and transition to middle school poses a risk for social challenges, question arises whether poor reading skills predict victimization and bullying across the elementary to middle school transition.

## The present study

The aim of the present study is to investigate cross-lagged associations between reading skills and involvement in bullying across the transition from elementary to middle school. We examine this separately for reading fluency and comprehension. Our goal is to investigate whether poor reading skills predict later involvement in school bullying, after controlling for externalizing and internalizing problems. The analysis is based on a simple model in which reading difficulties and externalizing/internalizing problems accumulate and form a risk of bullying involvement [35, 74–78], and also co-occur [51, 79]. Because externalizing and internalizing problems have been reported to be common among poor readers [e.g. 49–56], are among the strongest risk factors of bullying and victimization [80], respectively, and this increased risk is apparent also across the transition to middle school [81], they are taken into account in the analyses. Firstly, we are interested in the extent to which poor reading skills in the end of elementary school (Grade 6) increase the risk for victimization or bullying after transition to middle school (Grade 7). Moreover, we examine whether poor reading skills pose a risk for victimization or bullying during middle school, between Grade 7 and Grade 9. Based on the previous research, we hypothesize that poor reading skills predict later bullying perpetration (Hypothesis 1) and victimization (Hypothesis 2). These associations are predicted to be stronger across the transition from elementary to middle school (i.e., between Grades 6 and 7; Hypothesis 3) than during middle school. We hypothesize the associations to be similar for reading fluency and comprehension.

## Method

### Participants and procedure

The present data came from an extensive longitudinal age cohort study [82], in which a community sample of children ($n$ = 1,880) were followed from kindergarten entry (age $M$ = 74.0 ± 3.6 months) to the end of middle school (Grade 9, age 15). In Finland, nine years comprehensive school will start in the year children turn seven years of age. The transition from elementary to middle school takes place between Grades 6 and 7.

The study includes the entire age cohort in one rural municipality and two medium size towns, plus about a half of the age cohort from one big city during Grades 6, 7, and 9, in order

to follow them across the transition to middle school. The final sample consists of 1,824 students (47.3% female), whose parents gave an informed consent for their child to participate and who were still participating the study in Grade 6. Of these students, 1,715 (47.3% female) participated also in T2, and 1,647 (47.1% female) in T3. The longitudinal study [82] has been evaluated and approved by the ethics committee of the University of Jyväskylä (June 6, 2006). In the present study, we used data collected from the students reading fluency and comprehension tests instructed by trained research assistants, as well as questionnaires about externalizing/internalizing problems, bullying, and victimization during regular school lessons at Grade 6 (T1, April 2013), Grade 7 (T2, April 2014), and Grade 9 (T3, April 2016).

## Measures

**Reading fluency.**   There were two group-administered tests for the assessment of reading fluency in each grade (T1, T2, and T3): a word reading fluency task and a word-chain task. First, the word reading fluency task is a subtest of the nationally normed reading test battery [ALLU; 83]. In this speed test with a two-minute time limit, up to a maximum of 80 trials could be taken, each involving a picture and four phonologically similar words, with the task being to draw a line to match the picture with the semantically matching word. Form A of the original test was used in T1, and a similarly structured form B created for the purpose of the current study with phonologically more difficult words was used in middle school (T2 and T3). The score was the number of correct responses, reflecting both fluency in reading the stimulus words and accuracy in making a correct choice from the alternatives. The Kuder–Richardson reliability coefficient was .96 at T1, .93 at T2, and .94 at T3.

Second, a time-limited word-chain test was utilized to assess reading fluency. In T1 and T2, the test consists of silently reading a total of 10 rows of word-chains comprising four to six words written together, and marking the word boundaries with a vertical line [84]. The original test form was used in T1, and a similarly structured form created for the purpose of the current study with phonologically more difficult words was used in T2. The number of correctly drawn lines between words during one minute was calculated in each time point (maximum 40). In T3, similar test form from a different test battery for adolescents and adults was used [85]. There were 25 rows of word-chains and the time-limit was 1 min 30 s (maximum 75 points). The Kuder–Richardson reliability coefficient was .96 at T3, which was the only time point with item-level information in the data. The Pearson correlation coefficient was .65 between T1 and T2, .64 between T1 and T3, and .70 between T2 and T3.

The sums of the reading fluency tests were standardized and the total fluency score was calculated for each time point as an average of the two standardized sum scores. The Cronbach's alphas of the total scores were .69 at T1, .77 at T2, and .79 at T3.

**Reading comprehension.**   To assess reading comprehension in T1, a subtest of the nationally normed reading test battery [ALLU; 83] was used. In T2 and T3, a similar standardized reading comprehension test developed for middle school was utilized [YKÄ; 86]. All tests had the same aim, instruction, and number of multiple tasks, but different texts and items. The students were asked to silently read a non-fiction text. They answered 11 multiple-choice questions and one question in which they had to arrange five statements in the correct sequence based on the information gathered from the text. The text contained 557 words in T1, 452 in T2, and 500 in T3. Number of correct answers was calculated (maximum 12). Each participant completed the task at his or her own pace, but the maximum time allotted was 30 min. The Kuder–Richardson reliability coefficient was .65 at T1, .68 at T2, and .63 at T3.

**Self-reported bullying and victimization.**   Self-reported bullying and victimization were measured at each grade (T1, T2 and T3) with the global, single item bullying and victimization

questions from the revised Olweus Bully/Victim Questionnaire, OBVQ [87]. The global items have been shown to be valid measures of bullying and victimization [88], and they have also been used extensively in Finland [89, 90]. For example, the global victimization question correlates positively with the nine OBVQ items indicating specific forms of victimization [91], as well as with several expected correlates of victimization [92]. The students were explained the definition of bullying, emphasizing its repetitive nature and the power imbalance between bully and victim, and they could also read the definition on the self-report questionnaire. They were asked how often they *had been bullied*, and how often they *had bullied others* at school in the last couple of months. Answer was given on a 5-point scale (1 = not at all, 2 = only once or twice, 3 = two or three times a month, 4 = about once a week, and 5 = several times a week). Bullying and victimization were used as continuous variables in the analyses.

**Externalizing and internalizing problems.**   Externalizing and internalizing problems were measured at each grade (T1, T2, and T3) using the Finnish version of self-report form of the Strengths and Difficulties Questionnaire [SDQ; 93, 94], which has been shown to be a highly valid screening instrument [95], and to have good psychometric properties among Finnish children and adolescents [96, 97]. The SDQ consists of 25 items rated on a 3-point scale (i.e., 0 = not true; 1 = somewhat true; 2 = certainly true), producing scales for hyperactivity/inattention, conduct problems, emotional symptoms, peer problems, and prosociality. To measure *externalizing problems*, we used the scales for hyperactivity/inattention (five items, e.g., restless, cannot stay still for long) and conduct problems (five items, e.g., often fights with other children or bullies them). The composite score for externalizing problems for each grade was formed as the mean score of the hyperactivity/inattention and conduct problems scales. To measure *internalizing problems* we used with the emotional symptoms subscale (5 items) of the questionnaire.

The expected factor structures of the SDQ scales and subscales were tested with confirmatory factor analysis. For externalizing problems, a second-order confirmatory factor model was fitted to the data with hyperactivity and conduct problems factors (five items loading on each) loading on the higher-order externalizing problems factor. The five items of emotional symptoms subscale formed the internalizing problems factor. Residual correlations between the same items were allowed in T1, T2, and T3. The original model did not have an acceptable fit ($\chi^2$(874) = 2724.10, CFI = .90, TLI = .88, RMSEA = .03, SRMR = .06) and one item of the conduct problems subscale ("I usually do as I am told") had a very small factor loading (T1: .10, T2: .03, T3: .06). After omitting this item from the scale, the model fitted the data ($\chi^2$(750) = 2215.12, CFI = .91, TLI = .90, RMSEA = .03, SRMR = .05) and the results supported the distinctness of externalizing and internalizing (emotional) problems factors. For the lower level factors, the factor loadings ranged as follows: for hyperactivity from .30 to .67 in T1, from .28 to .71 in T2, and from .27 to .75 in T3, for conduct problems from .52 to .60 in T1, from .54 to .70 in T2, and from .57 to .65 in T3, for internalizing problems (emotional symptoms) from .41 to .72 in T1, from .49 to .71 in T2, and from .48 to .76 in T3. For the second order factor of externalizing problems, factor loadings ranged from .72 to .92 in T1, from .72 to .90 in T2, and from .71 to .85 in T3. Externalizing and internalizing problems factors correlated .54 in T1, .63 in T2, and .57 in T3.

Finally, mean scores for externalizing and internalizing problems were calculated at each time point. Externalizing problems subscale was computed as a mean score of 9 items on the hyperactivity and conduct problem scales ranging from 0 to 2 with an Omega (ω) of .73 at T1, .75 at T2, and .77 at T3. Internalizing problems subscale was computed as a mean score of items on the emotional problems scale ranging from 0 to 2 with an Omega (ω) of .70 at T1, .73 at T2, and .75 at T3.

## Analysis strategy

The aim of the present study was to investigate how poor reading skills are related to bullying and victimization across the transition from elementary to middle school, controlling for externalizing and internalizing problems. Two longitudinal cross-lagged panel models with 3 time-points (T1, T2, and T3) were fitted to the data separately for reading fluency (Model A) and comprehension (Model B), utilizing Mplus statistical package [Version 7.4; 87]. With cross-lagged panel model it is possible to estimate the directional influence variables have on each other over time, and thus examine the causal influences between variables [98]. This enabled us to test whether poor reading skills longitudinally predict victimization and bullying, and not the other way around.

For predictor variables in T1, the ICCs for classroom level were .13 for reading fluency, .08 for comprehension, .07 for victimization, .09 for bullying, .01 for externalizing problems, and .03 for internalizing problems. This indicates that some of the variability in T1 variables was due to differences between classrooms. However, because we were not interested in associations between classrooms, and because the design effects of the predictors varying between .15 (externalizing problems) and 1.40 (reading fluency) were below the threshold of 2 [99], multi-level modeling was not utilized. Instead, the differences between classrooms were taken into account by using COMPLEX option [100, 101] that estimates the model at the level of the whole sample but corrects for distortions in standard error estimates caused by the clustering of observations (i.e., between-level variation). In order to take into account the maximum amount of data, T1 classroom was used as a cluster. The variables at T1 were grand-mean centered in order to assess the regression coefficients relative to the population average [102]. The goodness of the fit was evaluated with the following absolute and relative fit indices [103]: 1) Chi-squared test ($\chi^2$/df < 2: good fit); 2) the Comparative Fit Index (CFI, >.90: acceptable; >.95: good fit); 3) the Tucker–Lewis Index (TLI, CFI, >.90: acceptable; >.95: good fit); 4) root mean square error of approximation (RMSEA, < .06: good fit; < .08 acceptable); and 5) standardized root mean square residual (SRMR, < .05: good fit; < .08 acceptable).

Missing data for T1 ranged from 0.02% (Reading skills) to 1.1% (Bullying), for T2 from 7.2% (Reading skills) to 8.7% (Externalizing and internalizing problems), and for T3 from 11.0% (Reading skills) to 11.8% (Externalizing and internalizing problems). Little's MCAR test showed that missing values were randomly distributed for all the variables in T1 ($\chi^2(19) = 23.71$, $p = .207$) and in T2 ($\chi^2(25) = 24.15$, $p = .511$), but not in T3 ($\chi^2(22) = 70.11$, $p < .001$). The students that no longer participated in T3 had weaker reading fluency ($t(1818) = 4.70$, $p < .001$, $d = .37$) and comprehension ($t(1819) = 4.22$, $p <. 001$, $d = .34$), more externalizing problems ($t(1080) = 3.34$, $p < .01$, $d = -.27$), and more bullying perpetration ($t(1802) = 3.12$, $p < .01$, $d = -.25$) in T1 compared to those that remained in the study until T3. There was no difference between those who remained and those who did not in victimization or internalizing problems. In order to use all available data to estimate the model without imputing them, full information maximum likelihood (FIML) estimation with robust standard errors was used to handle the missing data and to account for the non-normality of some of the study variables in the subsequent analyses [100].

## Results

The correlations and descriptive statistics of the study variables are presented in Table 1. Reading fluency and comprehension measures correlate moderately with each other at T1, T2, and T3, but negatively with bullying others, as well as with externalizing problems. However, the correlation between fluency and externalizing problems is only significant in middle school (T2 and T3), not in elementary school (T1). Reading measures also correlate weakly negatively

**Table 1. Intercorrelations and descriptive statistics of the study variables.**

| Variable | 1 | 2 | 3 | 4 | 5 | 6 | 7 | 8 | 9 | 10 | 11 | 12 | 13 | 14 | 15 | 16 | 17 | 18 | M | SD |
|---|---|---|---|---|---|---|---|---|---|---|---|---|---|---|---|---|---|---|---|---|
| 1. RF T1 | - | | | | | | | | | | | | | | | | | | 0.00 | 0.87 |
| 2. RF T2 | .73*** | - | | | | | | | | | | | | | | | | | 0.00 | 0.90 |
| 3. RF T3 | .72*** | .77*** | - | | | | | | | | | | | | | | | | -0.00 | 0.91 |
| 4. RC T1 | .31*** | .38*** | .38*** | - | | | | | | | | | | | | | | | 7.15 | 2.55 |
| 5. RC T2 | .27*** | .36*** | .37*** | .50*** | - | | | | | | | | | | | | | | 6.60 | 2.54 |
| 6. RC T3 | .27*** | .34*** | .37*** | .47*** | .51*** | - | | | | | | | | | | | | | 7.01 | 2.44 |
| 7. Vict T1 | -.04 | -.04 | -.02 | -.03 | -.00 | -.01 | - | | | | | | | | | | | | 1.47 | 0.93 |
| 8. Vict T2 | -.05* | -.06* | -.05* | -.05* | -.06* | -.01 | .38*** | - | | | | | | | | | | | 1.41 | 0.90 |
| 9. Vict T3 | -.03 | -.04 | -.11*** | -.04 | -.02 | -.06* | .15*** | .21*** | - | | | | | | | | | | 1.21 | 0.65 |
| 10. Bul T1 | -.06* | -.08*** | -.11*** | -.10*** | -.13*** | -.13*** | .27*** | .15*** | .08** | - | | | | | | | | | 1.31 | 0.62 |
| 11. Bul T2 | -.08*** | -.07** | -.11*** | -.12*** | -.15*** | -.14*** | .12*** | .22*** | .12*** | .33*** | - | | | | | | | | 1.29 | 0.65 |
| 12. Bul T3 | -.09*** | -.10*** | -.13*** | -.12*** | -.14*** | -.14*** | .05* | .06* | .38*** | .20*** | .30*** | - | | | | | | | 1.18 | 0.61 |
| 13. Ext T1 | -.03 | -.10*** | -.12*** | -.20*** | -.21*** | -.20*** | .16*** | .10*** | .10*** | .30*** | .22*** | .18*** | - | | | | | | 0.43 | 0.30 |
| 14. Ext T2 | -.03 | -.11*** | -.13*** | -.19*** | -.25*** | -.24*** | .12*** | .12*** | .14*** | .22*** | .25*** | .25*** | .59*** | - | | | | | 0.46 | 0.32 |
| 15. Ext T3 | -.06* | -.12*** | -.14*** | -.18*** | -.24*** | -.26*** | .06* | .08** | .19*** | .17*** | .18*** | .27*** | .48*** | .59*** | - | | | | 0.45 | 0.33 |
| 16. Int T1 | .04 | .01 | .05* | -.03 | -.02 | .03 | .30*** | .22*** | .09*** | .12*** | .05* | .03 | .40*** | .29*** | .20*** | - | | | 0.51 | 0.43 |
| 17. Int T2 | .04 | .01 | .03 | -.04 | -.06* | -.03 | .18*** | .27*** | .14*** | .06* | .10*** | .07** | .26*** | .44*** | .24*** | .55*** | - | | 0.52 | 0.44 |
| 18. Int T3 | .05* | .05* | .03 | .01 | .00 | .02 | .13*** | .17*** | .21*** | .01 | .02 | .05* | .20*** | .23*** | .38*** | .44*** | .53*** | - | 0.53 | 0.47 |

Note: RF = Reading fluency; RC = Reading comprehension; Vict = Victimization; Bul = Bullying; Ext = Externalizing problems; Int = Internalizing problems

$***p < .001$

$**p < .01$

$*p < .05.$

with victimization, especially in T3, but neither of them correlates with internalizing problems. Bullying and victimization variables correlate positively with each other, indicating that some students are both victimized and bully others (bully/victims). In addition, bullying others correlates relatively strongly and positively with externalizing problems and to a lesser extent with internalizing problems, and victimization correlates relatively strongly and positively with internalizing problems, but also with externalizing problems. Finally, externalizing and internalizing problems are intercorrelated. Mean scores suggest that mean levels of bullying and victimization are declining with age, especially from Grade 7 to Grade 9, but the levels of externalizing and internalizing problems stay approximately the same.

In the longitudinal cross-lagged panel model with 3 time-points (T1, T2, and T3), in addition to stability coefficients for all the constructs (reading fluency/comprehension, victimization, bullying, externalizing problems, and internalizing problems), all cross-lagged paths between the constructs were estimated. The constructs at T1, as well as the residuals of the constructs in T2 and T3 were allowed to correlate within each time point.

First, a cross-lagged panel model was fitted for reading fluency (Model A). Originally, the model did not fit the data ($\chi^2(25)$ = 319.72, CFI = .94, TLI = .78, RMSEA = .08, SRMR = .02). As suggested by the modification indices, autoregressive paths were added from T1 reading fluency to T3 reading fluency, from T1 externalizing problems to T3 externalizing problems, and from T1 internalizing problems to T3 internalizing problems. After these modifications the model fitted the data very well ($\chi^2(22)$ = 36.18, CFI = 1.00, TLI = .99, RMSEA = .02, SRMR = .01). This final model with the standardized beta coefficients is shown in Fig 1. Correlations between T1 variables, and residual correlations between T2 and T3 variables are presented in Table 2.

As depicted in Fig 1, in addition to strong autoregressive paths between reading fluency in different time points, fluency is longitudinally negatively associated with bullying others, both between Grades 6 and 7, and between Grades 7 and 9, so that students with poorer reading fluency report bullying others more at the later time point. Fluency is not longitudinally associated with victimization. The association between earlier fluency and later bullying perpetration is similar between Grades 6 and 7, as it is between Grades 7 and 9, and non-significant between fluency and victimization in both comparisons.

For bullying others, autoregressive paths between T1 and T2, as well as between T2 and T3 are approximately of the same magnitude and the unstandardized coefficients do not differ significantly ($\Delta b$ = -.06, $p$ = .274), but for victimization, the association between T1 and T2 is significantly stronger than between T2 and T3 ($\Delta b$ = -.19, $p$ < .001). Besides earlier bullying

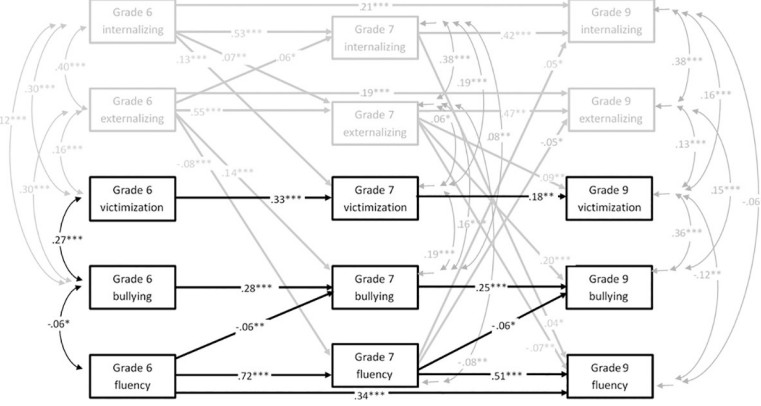

**Fig 1. Cross-lagged panel model for reading fluency (Model A), significant standardized coefficients.**

**Table 2. Correlations between T1 variables, and residual correlations between T2 and T3 variables (Model A).**

| Variable | 1 | 2 | 3 | 4 | 5 | 6 | 7 | 8 | 9 | 10 | 11 | 12 | 13 | 14 | 15 |
|---|---|---|---|---|---|---|---|---|---|---|---|---|---|---|---|
| 1. RF T1 | - | | | | | | | | | | | | | | |
| 2. Vict T1 | -.04 | - | | | | | | | | | | | | | |
| 3. Bul T1 | -.06* | .27*** | - | | | | | | | | | | | | |
| 4. Ext T1 | -.03 | .16*** | .30*** | - | | | | | | | | | | | |
| 5. Int T1 | .04 | .30*** | .12*** | .40*** | - | | | | | | | | | | |
| 6. RF T2 | | | | | | - | | | | | | | | | |
| 7. Vict T2 | | | | | | -.03 | - | | | | | | | | |
| 8. Bul T2 | | | | | | .01 | .19*** | - | | | | | | | |
| 9. Ext T2 | | | | | | -.08** | .06* | .16*** | - | | | | | | |
| 10. Int T2 | | | | | | -.01 | .19*** | .08** | .38*** | - | | | | | |
| 11. RF T3 | | | | | | | | | | | - | | | | |
| 12. Vict T3 | | | | | | | | | | | -.12** | - | | | |
| 13. Bul T3 | | | | | | | | | | | -.05 | .36*** | - | | |
| 14. Ext T3 | | | | | | | | | | | -.04 | .13*** | .15*** | - | |
| 15. Int T3 | | | | | | | | | | | -.06* | .16*** | .03 | .38*** | - |

Note: RF = Reading fluency; Vict = Victimization; Bul = Bullying; Ext = Externalizing problems; Int = Internalizing problems

*** $p < .001$

** $p < .01$

* $p < .05$.

perpetration, externalizing problems is the strongest predictor of later bullying perpetration. Internalizing problems predict victimization across the transition to middle school (from T1 to T2), but not during middle school (from T2 to T3). Regarding the covariates, Grade 7 (T2) fluency is related negatively with externalizing problems and positively with internalizing problems in Grade 9 (T3). On the other hand, externalizing problems relate to weaker fluency and more bullying perpetration later on.

Victimization and internalizing problems are positively related in each time point concurrently (See Fig 1 and Table 2), but victimization does not predict later internalizing problems. The same is true for victimization and externalizing problems, as well as victimization and bullying. Residuals of fluency are concurrently associated negatively with the residuals of internalizing problems in T2 and T3. In T3, the residuals of fluency and victimization are also negatively associated. The positive concurrent associations between the covariates (externalizing/internalizing problems) and bullying and victimization are significant and can be found in Fig 1.

Second, a cross-lagged panel model was fitted for reading comprehension (Model B). Again, the original model had some poor fit-indices ($\chi^2(25) = 233.87$, CFI = .95, TLI = .80,

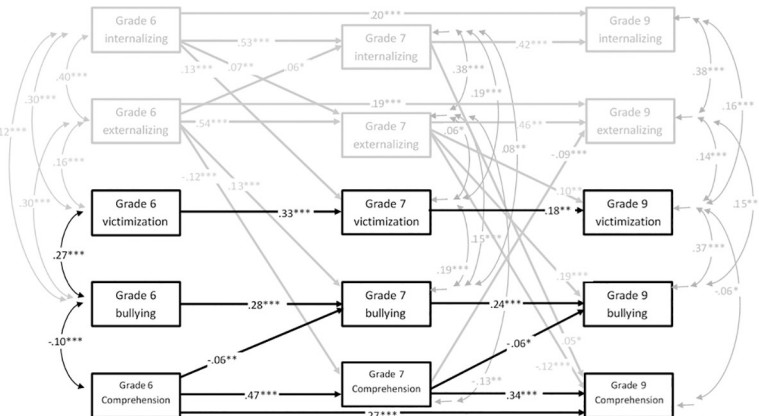

**Fig 2. Cross-lagged panel model for reading comprehension (Model B), significant standardized coefficients.**

RMSEA = .07, SRMR = .03). Parallel with Model A, adding autoregressive paths from T1 reading comprehension to T3 reading comprehension, from T1 externalizing problems to T3 externalizing problems, and from T1 internalizing problems to T3 internalizing problems resulted in a very well-fitting final model shown in Fig 2 with the standardized beta coefficients ($\chi^2$(22) = 33.68, CFI = 1.00, TLI = .99, RMSEA = .02, SRMR = .01). Correlations between T1 variables, and residual correlations between T2 and T3 variables are presented in Table 3.

As Fig 2 shows, autoregressive paths between reading comprehension in different time points are relatively strong, although not as strong as for fluency. Similarly with fluency, comprehension is longitudinally negatively associated with bullying others, both between Grades 6 and 7, and between Grades 7 and 9, but not with victimization. Grade 6 bullying is negatively

**Table 3. Correlations between T1 variables, and residual correlations between T2 and T3 variables (Model B).**

| Variable | 1 | 2 | 3 | 4 | 5 | 6 | 7 | 8 | 9 | 10 | 11 | 12 | 13 | 14 | 15 |
|---|---|---|---|---|---|---|---|---|---|---|---|---|---|---|---|
| 1. RC T1 | - | | | | | | | | | | | | | | |
| 2. Vict T1 | -.03 | - | | | | | | | | | | | | | |
| 3. Bul T1 | -.10* | .27*** | - | | | | | | | | | | | | |
| 4. Ext T1 | -.20 | .16*** | .30*** | - | | | | | | | | | | | |
| 5. Int T1 | .03 | .30*** | .12*** | .40*** | - | | | | | | | | | | |
| 6. RC T2 | | | | | | - | | | | | | | | | |
| 7. Vict T2 | | | | | | -.04 | - | | | | | | | | |
| 8. Bul T2 | | | | | | -.07** | .19*** | - | | | | | | | |
| 9. Ext T2 | | | | | | -.13*** | .06* | .15*** | - | | | | | | |
| 10. Int T2 | | | | | | -.05* | .19*** | .08** | .38*** | - | | | | | |
| 11. RC T3 | | | | | | | | | | | - | | | | |
| 12. Vict T3 | | | | | | | | | | | -.06* | - | | | |
| 13. Bul T3 | | | | | | | | | | | -.03 | .37*** | - | | |
| 14. Ext T3 | | | | | | | | | | | -.00 | .14*** | .15*** | - | |
| 15. Int T3 | | | | | | | | | | | -.09*** | .16*** | .03 | .38*** | - |

Note: RC = Reading comprehension; Vict = Victimization; Bul = Bullying; Ext = Externalizing problems; Int = Internalizing problems

***$p < .001$

**$p < .01$

*$p < .05$.

associated with comprehension in Grade 7. As regard to covariates, there is a reciprocal negative association between comprehension and externalizing problems both between Grades 6 and 7, and between Grades 7 and 9, but comprehension is not longitudinally associated with internalizing problems. Comprehension is concurrently negatively associated with externalizing problems in each time point, with bullying in T1 and T2, with victimization in T3, and also weakly with internalizing problems in T2 (see Fig 2 and Table 3). The longitudinal and concurrent relations between victimization, bullying, externalizing problems and internalizing problems are similar to Model A.

## Discussion

The aim of the present study was to investigate the social concomitants of poor reading ability across the transition from elementary to middle school, which takes place in Finland after Grade 6 (12 years of age). This study is the first one to explore associations between poor reading, bullying, and victimization in adolescence, and the first one to take into account reading comprehension when examining the associations with RDs and bullying involvement.

We found that fluency and comprehension relate to bullying involvement quite similarly. Both fluency and comprehension negatively predicted later bullying perpetration, both from Grade 6 to Grade 7 and from Grade 7 and Grade 9. Thus, Hypothesis 1 was confirmed. Reading skills were negatively associated with bullying perpetration, so that students with poorer reading skills reported bullying others more at the later time point, even when externalizing and internalizing problems were controlled for. This is in accordance with the conclusions of Turunen et al. [35] concerning younger elementary school students, that reading difficulties and externalizing/internalizing problems accumulate and form a risk of bullying perpetration. We replicated this finding with a large sample of adolescents and utilizing reading comprehension measures in addition to fluency measures. Our results support the notion that poor reading may trigger frustration and antisocial behavior also towards peers [35, 36]. On the other hand, since in adolescence bullying is also used as a strategy to increase dominance status [10, 22], this may be the motivation behind poor readers' bullying perpetration. RDs may be a factor threatening one's social status, and gaining social dominance by academic means may be difficult. Since SEN students worry more than their peers about being bullied in middle school [1], it is possible that also poor readers are particularly concerned about their position in the new group and compensate their worries by aggressive means, such as bullying others. However, it is important to remember that although the association turned consistent, it is small in magnitude. Not all struggling readers end up frustrated, aggressive, or bullying their peers.

Contrary to our expectation, neither fluency nor comprehension were longitudinally associated with victimization and Hypothesis 2 was rejected. In a previous cross-sectional study by Turunen et al. [36], self-reported reading difficulties were related especially to victimization, when difficulties in math and self-esteem were controlled for. Even though the authors controlled for self-esteem, the determining factor in their study may have been the personal experience of having a difficulty that was then depicted by peers as a vulnerability. In addition, several studies interviewing adolescents and adults with RDs about their experiences at school have retrospectively reported frequent victimization memories [i.e., 58, 59]. It is possible that even relatively sporadic events will be later remembered as continuous victimization experiences, or that other factors besides RDs explain these memories. Since there were no comparison groups in these qualitative studies, it is impossible to know whether students with RDs were victimized more often than their peers without difficulties. Nevertheless, we could not replicate these findings in our adolescent sample longitudinally with standardized reading fluency or comprehension test scores as indicators of RDs. This is in line with the research

showing that poor academic performance is a significant predictor of bullying behavior, but the same cannot be said about victimization [80]. It seems that, in adolescence, being a poor reader is not likely to make students easy targets for later victimization. However, reading skills and victimization were negatively associated concurrently in Grade 9 in the current study. The forthcoming transition to high school, and decisions and plans related to that (i.e., choosing academic versus vocational track), may demonstrate the difficulties to peers thus exposing struggling readers to victimization in the final year of compulsory education.

Finally, we predicted that reading skills would predict bullying and victimization more strongly across the transition from elementary to middle school (between Grades 6 and 7) than during middle school (between Grades 7 and 9). This was not the case, and Hypothesis 3 was rejected. For both measures of reading skills, the association with bullying was just as strong between Grades 6 and 7 as it was between Grades 7 and 9, and the association between reading skills and victimization was not significant in either comparison. The transition to middle school does not seem to add to the risk poor reading skills pose to bullying or victimization.

We replicated the previous robust findings that externalizing problems predict later bullying perpetration and internalizing problems predict, although less strongly, victimization [80]. Internalizing problems predicted victimization across the transition to middle school, but not during middle school. It seems that students with internalizing symptoms are at increased risk for being bullied upon entering middle school, but that risk abates in later adolescence. Moreover, victimization was concurrently associated with internalizing problems, but it did not predict later internalizing symptoms. Several studies have reported that victimized students exhibit concurrent and long-term psychological problems such as depression, anxiousness and psychosomatic symptoms [i.e. 20, 104–106]. In the present study, internalizing problems were operationalized as concurrent emotional symptoms (i.e., "I am often unhappy, depressed or tearful" or "I have many fears, I am easily scared"). It seems that in this age group, or at least within this relatively short timeframe of three years, students show these kinds of emotional symptoms when they are being bullied, but previous victimization does not predict emotional symptoms one or two years later.

There are three main limitations in this study. Firstly, due to attrition, the sample size decreased from Grade 6 to Grade 9. To take this into account, full information maximum likelihood (FIML) estimation with robust standard errors was used to handle the missing data in the cross-lagged panel model [100]. Second, bullying and victimization were self-reported and measured with single items. Alternatively, forms of bullying and victimization [87], or peer reports [107] could have been utilized. However, the global items have been shown to be valid measures of bullying and victimization in previous studies [88]. Also, we did not evaluate the levels of victimization and bullying, or the associations between poor reading and bullying or victimization during elementary school before the transition to middle school. Thus, we have no information about whether victimization or bullying levels decreased before Grade 6 as suggested by previous studies [9, 19–23], or whether poor reading predicts bullying perpetration (or victimization) differently during elementary school than across or after the transition to middle school.

## Conclusions, implications for practice, and future directions

This study expanded the earlier findings reporting bullying and victimization experiences of students with learning difficulties and academic challenges [47] by investigating bullying involvement among students with poor performance in both reading fluency and comprehension. More specifically, we studied the interplay between reading skills, victimization, and

bullying across the transition from elementary to middle school. In comparison to previous studies on bullying among students with RDs that relied on interviews [i.e., 58] self-reports of RDs [36], or focused on reading fluency in the beginning of elementary school [35], we utilized standardized tests of reading fluency and comprehension in a relatively large, longitudinal sample of adolescents. We found that poor reading skills, both fluency and comprehension, were longitudinally associated with bullying perpetration, but not with victimization, when externalizing and internalizing problems were controlled for. Fluency and comprehension relate to bullying involvement very similarly.

There are some practical implications that can be drawn from these results. Although it is troublesome that poor reading skills increase risk for bullying perpetration, it should be kept in mind that the effects seem to be relatively small and other factors such as externalizing problems increase the risk more than poor reading skills do. That said, poor reading may be one factor contributing to a vicious cycle likely to lead to generalized feelings of dissatisfaction with school. On the other hand, as Vaz et al. [69] point out, transition to middle school brings with it a potential new opportunity for schools to provide support to disadvantaged students, as they continue to be at a disadvantage after the transition. Therefore, supporting reading and reading motivation of these students is one tool for educators to invite these students to a school participation unclouded by social problems.

In the future, exploring whether there are differences between schools and classrooms in how RDs predispose to bullying involvement would be important. It is possible that in some schools or classrooms the risk is realized, whereas in others it is not [108]. Even though we did not find longitudinal associations between reading skills and victimization, differences between classrooms and schools are plausible even regarding this. Thus, what characteristics of schools and/or classrooms make the transition from elementary to middle school smooth and successful for social relations of students with RDs? Moreover, what characteristics increase the risk of social problems? Finally, since several studies have now revealed that poor reading skills comprise a risk for bullying involvement, the logical next step is to examine the effects of antibullying interventions among this subgroup of at-risk students, both in elementary and middle school.

## Supporting information

**S1 File.**
(SAV)

## Author Contributions

**Conceptualization:** Tiina Turunen, Elisa Poskiparta, Christina Salmivalli, Pekka Niemi.

**Formal analysis:** Tiina Turunen.

**Funding acquisition:** Christina Salmivalli, Marja-Kristiina Lerkkanen.

**Investigation:** Tiina Turunen.

**Methodology:** Tiina Turunen.

**Project administration:** Elisa Poskiparta, Marja-Kristiina Lerkkanen.

**Resources:** Christina Salmivalli, Marja-Kristiina Lerkkanen.

**Supervision:** Elisa Poskiparta, Christina Salmivalli, Pekka Niemi.

**Visualization:** Tiina Turunen.

**Writing – original draft:** Tiina Turunen.

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
