## [Decision Letter · Decision Letter 0]

9 Jul 2020

PONE-D-20-08042

Longitudinal associations between poor reading skills, bullying and victimization across the transition from elementary to middle school

PLOS ONE

Dear Ms. Turunen,

Thank you for submitting your manuscript to PLOS ONE. After careful consideration, we feel that it has merit but does not fully meet PLOS ONE’s publication criteria as it currently stands. Therefore, we invite you to submit a revised version of the manuscript that addresses the points raised during the review process.

We urge you to consider the comments carefully, understanding that they are offered in the spirit of constructive criticism. If applicable, we recommend that you deposit your laboratory protocols in protocols.io to enhance the reproducibility of your results. Protocols.io assigns your protocol its own identifier (DOI) so that it can be cited independently in the future. For instructions see: http://journals.plos.org/plosone/s/submission-guidelines#loc-laboratory-protocols

We look forward to receiving your revised manuscript.

Kind regards,

Viktor Burlaka, LMSW, PhD

Academic Editor

PLOS ONE

Reviewers' comments:

Reviewer's Responses to Questions

5. Review Comments to the Author

Reviewer #1: The introduction is generally well written. However, in the next section, I feel that the authors should be a bit more explicit about the link between RD and bullying and RD and victimization. Are there any theoretical support for this association? If so, that should be included.

I think "bullying and transition to middle school" should be streamlined and perhaps included in the introduction. This section has already been discussed explicitly in other studies.

Hypotheses 1 and 2 have already been tested in other studies. I think the authors should really focus their hypotheses on the relevance of reading difficulties and bullying and victimization.

Given the study was conducted in Finland, a bit more information in the Introduction about bullying and academic difficulties in Finland would be helpful.

The authors should provide a brief rationale for the analyses chosen for the study.

For the bullying and victimization questionnaire, have the instruments been validated in studies conducted in Finland? If so, that should be included.

The results are well presented.

I would recommend including implications for practice, as the findings of this study would have implications for practitioners and educators.

Reviewer #2: The manuscript addresses an important issue in exploring the potential risk factors for bullying and victimisation during adolescence, and specifically the longitudinal associations between reading difficulties and bullying perpetration and victimisation during the transition between schools and beyond. The researchers have employed a rigorous methodological approach, which utilises a large sample of children/adolescents, to address this issue. I have just a few suggested amendments below:

Introduction:

Line 58: Bullying is not always perpetrated against a less powerful person. Indeed recent definitions of bullying have highlighted the power imbalance involved can real, but also perceived.

Line 63-67: This paragraph is rather brief and seems a little disjointed from the surrounding information. Ideally the point(s) made in this paragraph could be expanded on or incorporated into other sections relating to internalising/externalising problems.

Line 104: ‘With regard to bullying’ opposed to ‘As regard to bullying’

Lines 108-110: It isn’t clear how the points made here relating to those with difficulties in dealing with worry worries are at greater risk also for social problems, are relevant to what is being addressed in this study. This would benefit from further elaboration or clearer linking.

Similarly lines 117-119: the relevance of the social context which is discussed; i.e., ‘those who remain victimized in social contexts where less victimization occurs experience more adjustment difficulties than those in contexts in which victimization is more common’, isn’t clear how it is relevant.

Line 169: This should read as either ‘less prepared for middle school’ or ‘less prepared in middle school’ (I seems as though an extra word has just been typed in here by mistake)

Discussion:

Lines 480-483: The discussion here relating to bullying being used in response to declining social status flowing the transition to a new school implies that perpetration increased slightly between grades 6 and 7, however this was not found. There are a couple of instances in the discussion where this is implied, and could therefore be re-word slightly to ensure greater clarity and constancy with the findings.

---

## [Author Response · Author response to Decision Letter 0]

23 Oct 2020

Reviewer #1: 

The introduction is generally well written. However, in the next section, I feel that the authors should be a bit more explicit about the link between RD and bullying and RD and victimization. Are there any theoretical support for this association? If so, that should be included.

- We now explain in more detail how reading difficulties and externalizing/internalizing problems accumulate, co-occur and form a risk of bullying involvement. Please see the Present study section, page 12. 

I think "bullying and transition to middle school" should be streamlined and perhaps included in the introduction. This section has already been discussed explicitly in other studies.

- We took away the section and included some parts of it to the beginning of the Introduction on Page 3. The introduction now better reflects our aims after eliminating previous Hypotheses 1 and 2, as the Reviewer suggested (see next point)

Hypotheses 1 and 2 have already been tested in other studies. I think the authors should really focus their hypotheses on the relevance of reading difficulties and bullying and victimization.

- We decided to follow this advice and focus on the hypotheses on the relationship between reading fluency/comprehension and bullying and victimization, since this clearly makes the study more coherent. Therefore, we eliminated the previous Hypotheses 1 and 2, as well as all the related texts in the introduction, analyses, results, and discussion.

Given the study was conducted in Finland, a bit more information in the Introduction about bullying and academic difficulties in Finland would be helpful.

- We added information about reading difficulties in Finland to the beginning of the Reading difficulties and bullying -section and information about development of bullying and victimization in Finland from elementary to middle school to the first page of the Introduction.

The authors should provide a brief rationale for the analyses chosen for the study.

- We provided a brief rationale about choosing cross-lagged panel model for the study. Please see the Analysis Strategy -section.

For the bullying and victimization questionnaire, have the instruments been validated in studies conducted in Finland? If so, that should be included.

- We added information about the use of the questions in earlier studies in Finland. Please see page 15.

The results are well presented. I would recommend including implications for practice, as the findings of this study would have implications for practitioners and educators.

- The earlier version already had a paragraph about practical implications. However, it had no heading and may therefore be easily missed. We modified the heading of the last section which now reads “Conclusions, implications for practice, and future directions”. We also modified the section.

Reviewer #2: 

The manuscript addresses an important issue in exploring the potential risk factors for bullying and victimisation during adolescence, and specifically the longitudinal associations between reading difficulties and bullying perpetration and victimisation during the transition between schools and beyond. The researchers have employed a rigorous methodological approach, which utilises a large sample of children/adolescents, to address this issue. I have just a few suggested amendments below:

Introduction:

Line 58: Bullying is not always perpetrated against a less powerful person. Indeed recent definitions of bullying have highlighted the power imbalance involved can real, but also perceived.

- Following the suggestion of Reviewer 1, we have omittedthe previous Hypotheses 1 and 2 with related material in the Introduction. We have moved the definition of bullying to the first page of the Introduction. We modified the definition according to this comment to acknowledge that the power imbalance can be real, but also perceived.

Line 63-67: This paragraph is rather brief and seems a little disjointed from the surrounding information. Ideally the point(s) made in this paragraph could be expanded on or incorporated into other sections relating to internalising/externalising problems.

- We edited the paragraph and put it together with the following paragraph.

Line 104: ‘With regard to bullying’ opposed to ‘As regard to bullying’

- This was deleted after we streamlined the section as Reviewer #1 suggested and moved the rest of the sentence to the beginning of the introduction

Lines 108-110: It isn’t clear how the points made here relating to those with difficulties in dealing with worry worries are at greater risk also for social problems, are relevant to what is being addressed in this study. This would benefit from further elaboration or clearer linking.

- This was deleted after we streamlined the whole section as Reviewer #1 suggested

Similarly lines 117-119: the relevance of the social context which is discussed; i.e., ‘those who remain victimized in social contexts where less victimization occurs experience more adjustment difficulties than those in contexts in which victimization is more common’, isn’t clear how it is relevant.

- This was deleted after we streamlined the whole section as Reviewer #1 suggested

Line 169: This should read as either ‘less prepared for middle school’ or ‘less prepared in middle school’ (I seems as though an extra word has just been typed in here by mistake)

- This typo was corrected.

Discussion:

Lines 480-483: The discussion here relating to bullying being used in response to declining social status flowing the transition to a new school implies that perpetration increased slightly between grades 6 and 7, however this was not found. There are a couple of instances in the discussion where this is implied, and could therefore be re-word slightly to ensure greater clarity and constancy with the findings.

- This was deleted after we eliminated previous Hypotheses 1 and 2 as Reviewer #1 suggested.

---

## [Decision Letter · Decision Letter 1]

12 Mar 2021

Longitudinal associations between poor reading skills, bullying and victimization across the transition from elementary to middle school

PONE-D-20-08042R1

Dear Dr. Turunen,

We’re pleased to inform you that your manuscript has been judged scientifically suitable for publication and will be formally accepted for publication once it meets all outstanding technical requirements.

Kind regards,

Andrea Martinuzzi

Academic Editor

PLOS ONE

Additional Editor Comments (optional):

Reviewers' comments:

Reviewer's Responses to Questions

**Comments to the Author**

1. If the authors have adequately addressed your comments raised in a previous round of review and you feel that this manuscript is now acceptable for publication, you may indicate that here to bypass the “Comments to the Author” section, enter your conflict of interest statement in the “Confidential to Editor” section, and submit your "Accept" recommendation.

Reviewer #1: All comments have been addressed

Reviewer #2: All comments have been addressed

2. Is the manuscript technically sound, and do the data support the conclusions?

Reviewer #1: Yes

Reviewer #2: Yes

3. Has the statistical analysis been performed appropriately and rigorously? 

Reviewer #1: Yes

Reviewer #2: Yes

4. Have the authors made all data underlying the findings in their manuscript fully available?

Reviewer #1: Yes

Reviewer #2: Yes

5. Is the manuscript presented in an intelligible fashion and written in standard English?

Reviewer #1: Yes

Reviewer #2: Yes

6. Review Comments to the Author

Reviewer #1: Thank you for attending to the reviewers' comments. This is a significantly improved manuscript and should be accepted for publication.

Reviewer #2: I reiterate my initial comments that this manuscript addresses an important issue in exploring a potential risk factor for bullying and victimisation during adolescence. The researchers have recruited a large sample of children and adolescents to explore the longitudinal associations between reading difficulties and bullying perpetration and victimisation. The research is also rigorous and intricate with regards to its methodological approach and data analyses.

The manuscript has been further strengthened in the revision and the amendments made have addressed the initial comments from myself and the other reviewer. Aside from a final proof-read, I have no further required or suggested changes.

7. PLOS authors have the option to publish the peer review history of their article (what does this mean?). If published, this will include your full peer review and any attached files.

Reviewer #1: No

Reviewer #2: No

---

## [Editor Report · Acceptance letter]

22 Mar 2021

PONE-D-20-08042R1 

Longitudinal associations between poor reading skills, bullying and victimization across the transition from elementary to middle school 

Dear Dr. Turunen:

I'm pleased to inform you that your manuscript has been deemed suitable for publication in PLOS ONE. Congratulations! Your manuscript is now with our production department. 

Kind regards, 

on behalf of

Dr. Andrea Martinuzzi 

Academic Editor

PLOS ONE